

# The Mediterranean Outflow in the Strait of Gibraltar and its connection with upstream conditions in the Alborán Sea.

Jesús García-Lafuente[1], Cristina Naranjo[1], Simone Sammartino[1], José C. Sánchez-Garrido[1], Javier Delgado[1]

[1] Physical Oceanography Group, Department of Applied Physics 2, University of Málaga, Spain

*Correspondence to*: Jesús García Lafuente (glanfuente@ctima.uma.es)

**Abstract.** The present study addresses the hypothesis that the Western Alborán Gyre in the Alborán Sea (the westernmost Mediterranean basin adjacent to the Strait of Gibraltar) influences the composition of the outflow through the Strait of

Gibraltar. The process invoked is that strong and well-developed gyres help to evacuate the Western Mediterranean Deep Water from the Alborán basin, thus increasing its presence in the outflow, whereas weak gyres facilitates the outflow of Levantine and other Intermediate waters. To this aim, *in situ* observations collected at Camarinal (the main) and Espartel (the westernmost) sills of the Strait have been analysed along with altimetry data, which were employed to obtain a proxy of the strength of the gyre. An encouraging correlation of the expected sign was observed between the time series of potential

temperature at Espartel sill, which is shown to keep information on the outflow composition, and the proxy of the Western Alborán Gyre, suggesting the correctness of the hypothesis, although the weakness of the involved signals does not allow for drawing definitive conclusions.

## 1 Introduction

The Mediterranean outflow through the Strait of Gibraltar can be considered as formed by up to four water masses of

different origins (see Naranjo et al. (2015) for a recent analysis of the outflow composition). Of all the four, the Levantine Intermediate Water (LIW) characterized by an absolute maximum of salinity, and the Western Mediterranean Deep Water (WMDW), which is the densest and, often, the coldest water, have been historically seen as the main contributors to the outflow. The two remaining water masses are the Winter Intermediate Water (WIW) and the Tyrrhenian Dense Water (TDW), both of intermediate nature. The WIW is formed along the continental shelf of the Liguro-Provençal sub-basin and

Catalan Sea (Conan & Millot, 1995; Vargas-Yáñez, et al., 2012), exhibits marked interannual fluctuations that includes years of no formation (Pinot et al., 2002; Monserrat et al., 2008) and is characterized by an absolute minimum of potential temperature. Its volume transport is much less than the LIW and it flows embedded inside this water mass at relatively shallow depths. The TDW is the result of mixing of old WMDW residing in the Tyrrhenian Sea and newly entered LIW flowing into the western Mediterranean Sea through the Strait of Sicily (Rhein et al., 1999; Millot et al., 2006). It is slightly

denser than the LIW but lighter than the WMDW and spreads between both of them.




In the Alborán Sea, which is the upstream basin for the outflow, these waters approach the Strait following different paths, the LIW and WIW flowing closer to the Spanish continental slope and the WMDW attached to the Moroccan coast (Bryden and Stommel, 1982; Parrilla et al., 1986; Millot, 2009; Naranjo et al., 2012). This spatial differentiation is still observed before they flow over Camarinal sill (CS hereinafter, see Fig. 1), in the eastern half of the Strait (García-Lafuente et al., 2000; Naranjo et al., 2012, 2015). The severe mixing and dissipation that takes place in the Tangier Basin downstream (oceanwards) of CS (Wesson and Gregg, 1994; García Lafuente et al., 2009, 2011; Sánchez-Garrido et al., 2011), blurs this spatial pattern and tends to form a rather mixed outflow with small participation of entrained North Atlantic Central water in which the Mediterranean waters are barely distinguishable (García-Lafuente, et al., 2011; Naranjo et al., 2015).

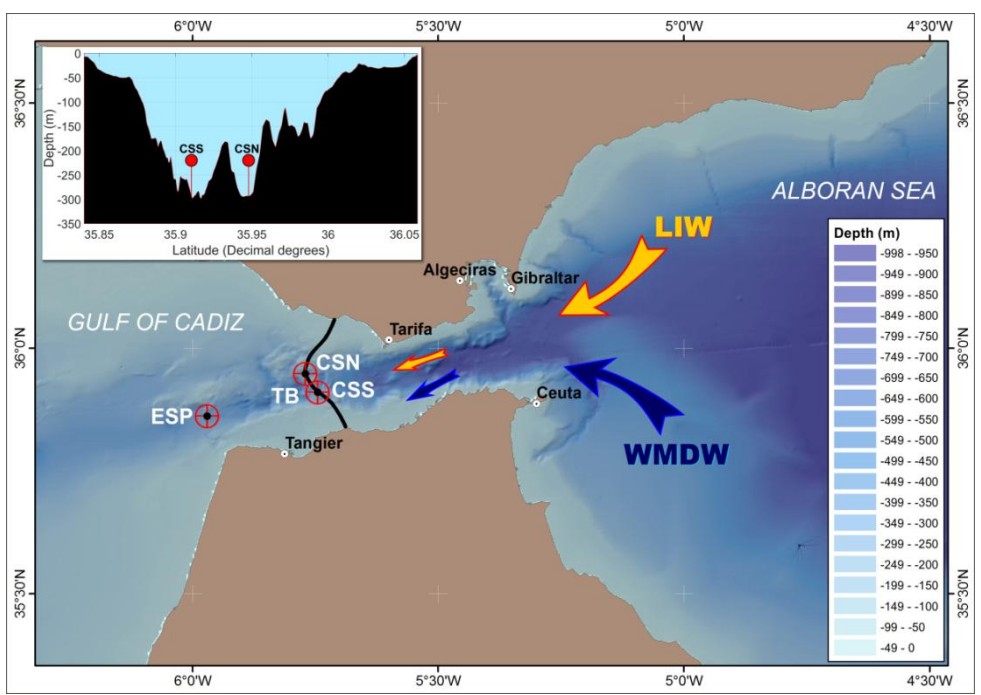

**Figure 1: Map of the Strait of Gibraltar and westernmost area of the Alborán Sea. The inset on the upper left corner shows the bathymetry of the Camarinal Sill (CS) section (solid black line in the map) and the location of the two mooring lines deployed in the north and south channels of the section (sites CSN and CSS, respectively). The location of the Espartel Sill long-term monitoring station of the outflow (ESP) and the Tangier Basin (TB) between both sills are also indicated. Coloured arrows sketch the main path of LIW and WMDW in the Alborán Sea and in the eastern half of the Strait.**

The bottom topography of the CS section depicts two troughs (Fig. 1). According to the spatial differentiation in the Alborán Sea and east of CS, the LIW, WIW (and TDW partially) will flow preferably through the northern channel of CS (CSN hereinafter, see Fig. 1), and the WMDW (and TDW partially as well) will do across the southern channel (CSS hereinafter). On the other hand, bearing in mind the different paths they follow in the Alborán Sea, the WMDW flow would benefit from a well-developed Western Alborán Gyre (WAG), a recurrent mesoscale feature of this sub-basin (Parrilla et al., 1986; Viudez et al., 1998), whereas the LIW and also the WIW would be favoured by a weak or absent WAG. While the first



process has been dealt with by different authors (Bryden and Stommel, 1982; Naranjo et al., 2012), the influence of the WAG on the LIW+WIW flow seems to not have been addressed yet.

The hypothesis of the present study is that if the WAG has an effect on the evacuation of these water masses, one expected consequence at CS section would be the variation of the outflow size across each channel, the presence of a well-developed

WAG increasing the size of the flow across CSS with regards to CSN, and *vice-versa*. Two mooring lines equipped with autonomous Conductivity-Temperature (CT) and uplooking Acoustic Doppler Current Profilers (ADCP) were deployed in these channels (Fig. 1) in order to get information about the characteristics and properties of the outflow at CS section. Since the flow through each channel advects water of different characteristics that mix downstream of CS, changing fractions of these flows can leave a footprint in the hydrological characteristics of the resulting mixed water that courses towards the

Atlantic Ocean. This outflow is being monitored by a long-term monitoring station deployed few tens of km westwards of CS at Espartel sill in year 2004 (ESP hereinafter, Fig. 1; see Sammartino et al., (2015) for a complete description of this station) and the collected long time series have been analysed to investigate our hypothesis as well. These observations have been complemented with altimetry data that have been employed to define a proxy of the WAG strength for checking the hypothesized WAG influence on the outflow composition. Regarding the objective of this paper, each set of the available *in*

*situ* observations has its own pros and cons. The instruments deployed at CSN and CSS sample Mediterranean waters that have not undergone important mixing yet, which is a clear pro for their identification, and the time series they collect (described in more detail in next Section) are well suited to investigate the spatial structure of the outflow through CS. Both aspects are of great interest to our study. However, taking into account that the timescale of the WAG variability is of the order of weeks to months (Vargas et al., 2002; Sánchez-Garrido et al., 2013), these series are too short to draw unarguable

conclusions about our hypothesis. This drawback does not apply to the long series at ESP, which shows up other disadvantages: since the characteristics of the Mediterranean waters flowing through CSN and CSS are not very different and the changing fractions that the WAG fluctuations can originate on these flows are also small, the footprint left by these fluctuations in the series collected at ESP are expectedly weak. This remark is of concern because other processes taking place in the Mediterranean Sea produce signals that, when exported through the Strait, can mask the weaker ones associated

with the WAG fluctuations. One of them is, for instance, the formation of larger-than-average volumes of denser-than-average WMDW in winter, as happened in the years of 2005 and 2006, which can uplift old WMDW nearby the Strait and facilitate its drainage rather independently of the WAG features, causing a cold signature in the temperature series at ESP (García-Lafuente et al., 2007). The interannual variability of the Mediterranean water properties may also interfere with the mechanism we propose. Last but not least, the entrained North Atlantic Central water that has been incorporated to the

outflow at this site, despite being a very small fraction (García-Lafuente et al., 2011), also contributes to blur possible signals (see Appendix A).

Despite the aforementioned sources of "unwanted noise" and observational limitations, our analysis provides reasonable evidence to support the feasibility of the hypothesis, even if it cannot be validated indisputably. The main goal of this paper is, therefore, to present and discuss those evidences bearing in mind these limitations, and has been organized as follows:



Next section describes the experimental data and the data processing, and presents some features visible from the processed data. Of particular interest is the corroboration of the aforementioned north-south spatial distribution of LIW/WMDW through CSN/CSS channels. Section 3, which is divided into two subsections, presents the results of the data analysis. The first subsection analyses the three-month field experiment carried out in Camarinal Sill, while the second one makes use of the multiyear time series collected at Espartel Sill to test our hypothesis. Finally, Section 4 discusses our findings and summarizes our conclusions.

## 2. Data and data-processing

### 2.1 Observations

#### 2.1.1 In situ Observations

Two twin mooring lines equipped with a CT probe and an uplooking ADCP located at 10m and 12m above the seafloor, respectively, were deployed in CSN and CSS channels at CS section (Fig. 1) in bottom depths of 306m and 310m, respectively. The ADCP observations span the period between 9 June and 25 September 2013, with a sampling interval of 4 min. The instruments were configured to sample 40 vertical bins 6m thick each, so that the velocity profile does not reach the sea surface although it covers the depth range of the outflow. The CT observations at CSN and CSS were made every 2 minutes and also started on 9 June, but finished earlier due to battery run outs (on 24 August and 27 August at CSN and CSS, respectively)

The historical time series of the outflow properties collected by the monitoring station deployed at ESP at 360m depth started in year 2004 and continues to the present. The instrument configuration of the monitoring station is similar to that of the lines deployed at CS although it contains additional probes to measure pH and $CO_2$ (Flecha et al., 2015), which are not used here. More information about the station can be found in Naranjo et al. (2015), or Sammartino et al. (2015). As seen below, only the potential temperature measured at around 15m above the seafloor from October 2004 to October 2015, sampled every 30 minutes, has been used in the present study.

The conductivity and temperature sensor of all the CT probes has an accuracy of $\pm 3 \times 10^{-3}$ mScm$^{-1}$ at 12°C and $\pm 2 \times 10^{-4}$ °C respectively, while the precision is $1 \times 10^{-4}$ mScm$^{-1}$ for the conductivity and $1 \times 10^{-4}$ °C for the temperature. The horizontal velocity measured by the ADCPs of the twin moorings of the CS has a resolution of 1mm/s while its accuracy is 1cm/s.



### 2.1.2 Altimetry Data

Sea level anomaly (SLA) data have been downloaded from CMEMS ("Mediterranean sea L4 gridded maps rep SLA" from Copernicus Marine Environment Monitoring Service, http://marine.copernicus.eu), formerly distributed by AVISO. The product used here is a reanalysis of multi-altimeter satellite where sea surface heights were computed with respect to a twenty year mean and optimally interpolated to finally supply a gridded product with horizontal resolution of 1/8 degree in the Mediterranean Sea and with daily temporal resolution.

### 2.2 The processed time series

### 2.2.1 Temperature and Salinity

Figure 2 illustrates the typical strong tidal-induced variability of local temperature and salinity registered at the depths of the CT probes in CSN, CSS and ESP. Such variability masks the characteristics that the different outflowing water masses would have if tides were absent. A procedure to obtain a realistic estimation of such characteristics is to select the coldest or saltiest sample registered during each semidiurnal tidal cycle, as in García-Lafuente et al. (2009). With regards to CS section and under the hypothesis that LIW flows preferably through CSN, the saltiest option would be the right choice to select the representative long-term there, whereas the coldest option is more adequate to select samples in CSS, since WMDW has preference to flow attached to the Moroccan coast. For comparison purposes, however, the criterion must be the same, in which case the best option is to pick up the densest sample.





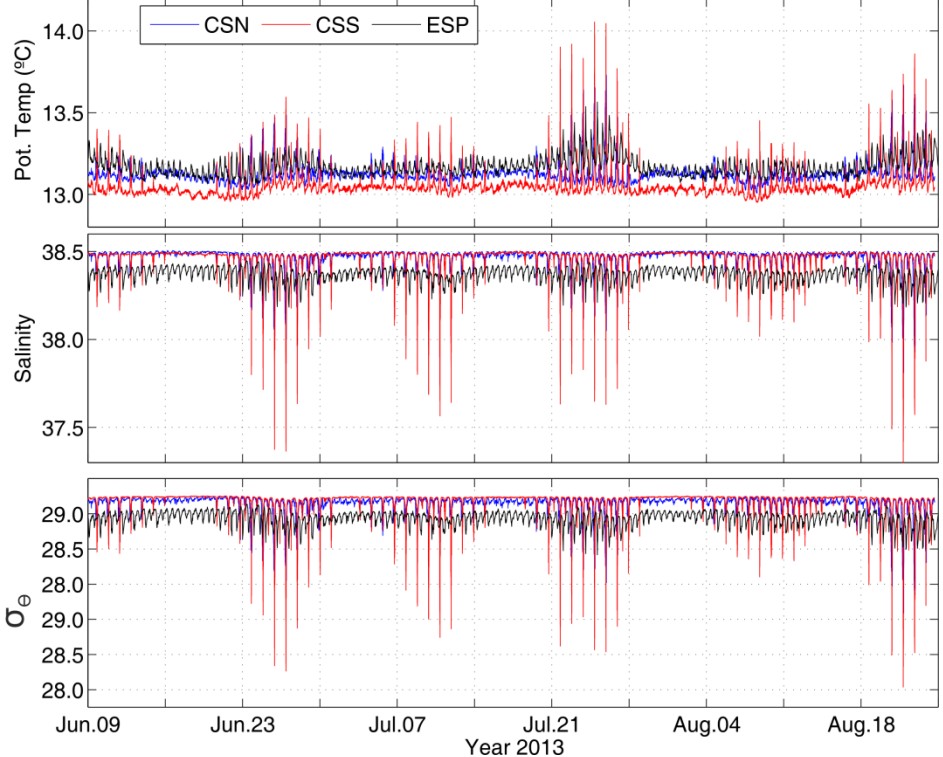

**Figure 2: Time series of potential temperature, salinity and σθ collected at CSN, CSS, and ESP sills (see legend). The strong fluctuations correspond to periods of spring tide and are greater in CSS, then in CSN, being the smallest in ESP. A noteworthy fact is the similitude of CSN and ESP temperature (CSS is clearly cooler) and of CSN and CSS salinity (ESP is clearly fresher).**

These samples are systematically observed near high water (Fig. 3) a fact that supports the procedure for selecting the most representative Mediterranean water samples: the barotropic tide in the Strait behaves like a standing wave (García-Lafuente et al., 1990; Candela et al., 1990) with the current heading west during the rising tide (flood) and east during the falling tide (ebb). Therefore, it is at the end of the flood that the water parcels located farther east in the Mediterranean side of the Strait

10 can reach and overflow the sill. Moreover they will be the least modified parcels by tidal mixing, which otherwise becomes more apparent over the sill near low water when the ebb current may reverse the "mean" flow and bring back to the Mediterranean Sea the water intensely mixed westwards (downstream) of CS (Wesson and Gregg, 1991; Sánchez-Garrido et al., 2013).





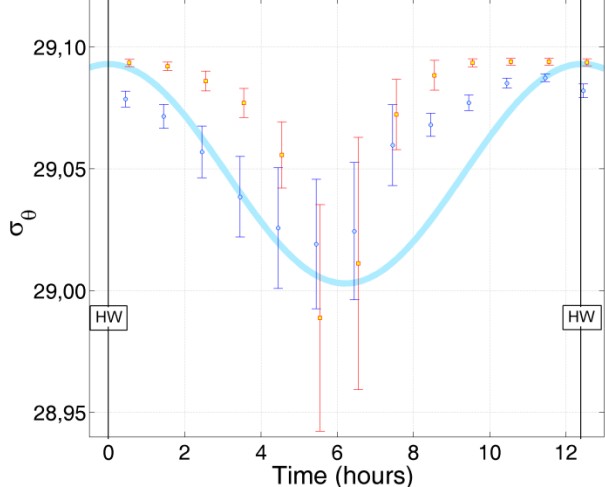

**Figure 3: Mean $\sigma_\theta$ values at CSN (blue circles) and at CSS (red squares) and standard deviation (bars) of all observations grouped in one-hour length bins. The curve refers to the sea level oscillation at Tarifa (see Fig.1 for location) and HW stands for High Water. The first bin includes all samples collected between HW and HW+1 hour regardless the date, the second one the samples between HW+1 and HW+2, and so on. The highest σθ values (which also have minimum std) are regularly observed near HW, whereas the lowest values spread around low water and show a marked variability.**

The result of this selection is displayed in Fig. 4. Figure 4A shows the TS diagram of the whole set of recorded data. Several historical CTD casts collected at ESP are also displayed to illustrate the mixing between Mediterranean and North Atlantic Central waters (NACW). The inset enlarges the area inside the black rectangle to show the subsets of data selected by the maximum-density procedure. Despite some overlapping of CSN and CSS data subsets, they are visually and statistically differentiable, confirming the hypothesis that LIW (and eventually other intermediate waters) flows out more attached to the northern half of CS in the bottom layer and the WMDW to the southern half. CTD data from ESP display the result of the mixing of waters flowing out through CSN and CSS with each other and with the overlying NACW, and how the differentiation observed at CS has already vanished.





**Figure 4: A) TS diagram of all samples recorded at CSN (blue), CSS (red), and ESP (black). Historical CTD casts collected at ESP site are displayed in light-blue dots. The typical linear temperature – salinity relationship is sketched by the double-headed arrow. The inset zooms on the area inside the black rectangle to show the subset of maximum density samples at each site (filled circles with the same colour code). Large filled circles indicate the mean values, which are also specified in the text boxes. B) Time evolution of potential temperature and salinity of the subset of maximum density (coloured dots) and a two-day low-pass filtered version of them (solid lines). Top panel is the sea level at Tarifa, which is included to illustrate the fortnightly cycle of the variables, more visible in the salinity series.**



### 2.2.2 Outflow per unit width

The ADCP observations have been integrated from the seafloor to the depth of the interface in order to obtain an estimate of the outflow per unit width across CSN and CSS channels. The velocity in the bottom layer has been extrapolated by fitting a logarithmic law on the deeper ADCP bins, from the depth of the deepest velocity to the seafloor where the velocity is set to zero. The most sensitive issue is the choice of the interface depth, which stems from the regular reversals of the tidal currents in CS (Candela et al., 1990; Bryden et al., 1994; García-Lafuente et al., 2000). Under these circumstances the best option can possibly be the selection of a material surface, usually an isohaline close to 37.3-37.5 (Bryden et al., 1994; García-Lafuente et al., 2000; Naranjo et al., 2014). However, this choice cannot be applied because of the lack of salinity profiles. An alternative solution, adopted here, is to take the depth of maximum velocity shear as the interface, as in Sánchez-Román et al. (2009) or Sammartino et al. (2015). In order to deal with the possible spikes in the ADCP velocity record, prior to compute flows, the current profiles have been smoothed vertically with a cubic polynomial fit with 1 meter resolution and temporally with a 20 minute moving average.

The computed flows per unit width are dominated by tides (Fig. 5A), with values ranging from $-433 m^2 s^{-1}$ to $+348 m^2 s^{-1}$ (mean value $-118 m^2 s^{-1}$) in CSN and $-499 m^2 s^{-1}$ to $+253 m^2 s^{-1}$ (mean value of $-132 m^2 s^{-1}$) in CSS, the minus sign indicating flow towards the Atlantic Ocean. In a first step, this high frequency tidal variability has been removed with a low-pass filter of $3 day^{-1}$ cut off frequency and the ensuing series are plotted in Fig. 5B.

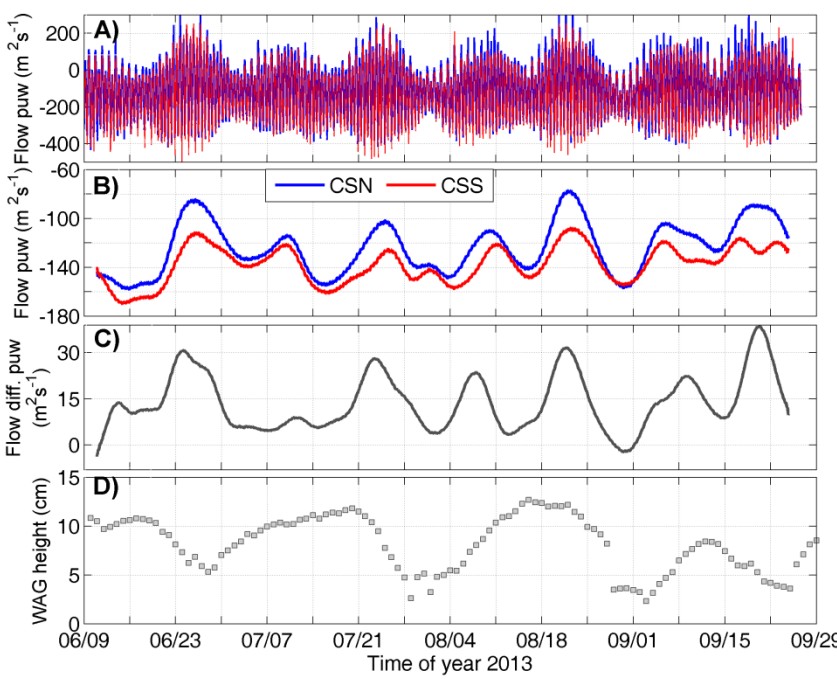

**Figure 5: A) Instantaneous flows per unit width across CSN and CSS (see legend). Negative sign indicates flow towards the Atlantic Ocean (outflow). B) Low-passed flows per unit width. C) Low-passed flow difference per unit width, computed as CSN-CSS from panel B. This difference is positive whenever the outflow through CSS is greater than through CSN, since both flows are negative. D) Daily mean sea surface height of the WAG (see text for details).**



### 2.2.3 Western Alborán Gyre

Altimetry data were processed to extract a proxy of the WAG strength. To this aim, the location of maximum sea level anomaly within the Western Alborán basin has been identified as the centre of the WAG. Starting from this point, the first local minimum of sea level anomaly in the northward direction has been found and the sea level difference between both points has been considered as the "height" of the gyre in that direction. The same procedure has been applied to the three other cardinal directions, and the averaged value of the four height anomalies has been adopted as the proxy of the WAG strength. We have also tried the mean sea surface gradient of the WAG, computed by dividing the height by the mean radius of the gyre, as an alternate proxy. The radius was estimated as the average distance from the centre of the WAG to the location of the local minimum in the four cardinal directions. The relatively coarse horizontal resolution of AVISO data produces a pronounced step-like series of the mean radius that does not change the pattern depicted by the height series too much, but nevertheless introduces noise in the gradient series. Therefore we opted for the first proxy, namely the height of the WAG, which is plotted in Fig. 5D.

### 3. Results

### 3.1 The observations at Camarinal sill and the Western Alborán Gyre

The panels of Fig. 4B show that the procedure of selecting the maximum density samples does not remove the fortnightly tidal variability in the series of temperature and salinity. The fortnightly cycle makes the flow through both channels be fresher and colder during spring tide and saltier and warmer during neap tides. Since the temperature and the salinity act in opposite directions, the density series show smoothed fortnightly fluctuations, though it still shows tendency to be lighter in spring and denser in neap tides, particularly at CSS (results not shown). These fluctuations are explained in terms of local enhanced (spring) or reduced (neap) mixing with the overlying NACW (Candela et al., 1990; Bryden et al., 1994; García-Lafuente et al., 2000) and with higher (lower) WMDW aspiration driven by the enhanced (reduced) tidal currents over the sill in spring (neap) tides to a lesser extent (Kinder and Bryden, 1990). Such variability performs as an inconvenient noise to detect signals that could come from the upstream basin carrying information about low-frequency variations in the Mediterranean.

A similar fortnightly variability is seen in the filtered series of flow per unit width through CSN and CSS (Fig. 5B). The flows are weaker in spring tides, a fact that has been ascribed to the aforementioned increase of the mixing during this phase of the cycle which reduces the effective density contrast between the exchanged flows, that is, the driving force of the exchange through the Strait (Bryden et al., 1994; García-Lafuente et al., 2000). Both subtidal flows are highly correlated (R=0.92 at 95% confidence level), indicating a coherent cross-strait structure of the velocity field. Nevertheless, there are small, though noteworthy, dissimilarities, which are just the features of interest and which are better seen in the series of the





difference of low-passed flows presented in Fig. 5C. It shows that the flow (in absolute value, that is, the outflow) is greater through the southern channel all the time except for a few days by the end of August and beginning of September when the flow difference is negative, indicating greater outflow through CSN (see also Fig. 5B).

This flow difference has been compared with the series of the proxy of the WAG (Fig. 5D). The latter is not expected to

exhibit much variability at short time scales due to the limited time resolution of the altimetry products. Even so, Fig. 5D shows several minima, two of which are particularly noticeable due to the marked decrease of the proxy from relatively high values: the first one happens at the beginning of August and the second one by the end of this month and beginning of September. They would point at a weakening or, even, a disappearance of the WAG, particularly during the second event when SLA-AVISO maps show no signs of WAG at all. The flow difference shows minima in both cases, the second one

being the situation of greater outflow through CSN mentioned above (Fig. 5C). Notice also that the two height maxima by the second half of July and August in Fig. 5D (stronger WAG) are roughly coincidental with similar local maxima of the flow difference (enhanced flow across CSS). All this agrees with the hypothesis that a weak WAG facilitates the outflow of LIW through CSN, while a strong WAG does the same with WMDW through CSS. However, there are several notorious discrepancies (the minima flow differences by the middle of August and September, for instance) that do not allow for

drawing unquestioned conclusions. A reason for this drawback could be the expected small size of the signal induced by WAG fluctuations on the flow structure across CS, which moreover is contaminated by the stronger fortnightly signal (Fig. 5B, 5C) and other energetic processes taking place in the Strait. Should the hypothesized cause-effect be a physical process, as the previous analysis partially supports, its identification with some degree of reliability would require significantly longer time series like those available at ESP.

**3.2 The observations at Espartel sill and the Western Alborán Gyre**

The time series of salinity and temperature in CSN and CSS may not be particularly sensitive to the WAG fluctuations, even if the flows per unit width are, because the flows would still carry the same type of water. The water properties at ESP, however, are the result of the strong mixing that takes place downstream of CS in the Tangier Basin, which in a simple model would be given by

$$X_{ESP} = \alpha_{CSN}X_{CSN} + \alpha_{CSS}X_{CSS} + \alpha_{NAC}X_{NAC} \qquad (1)$$

where $X$ denotes either salinity or temperature, $\alpha$ is the fraction of water involved in the mixture, subscripts CSN and CSS refer to water flowing through CSN and CSS channels, which would have properties close to LIW and WMDW, respectively. NAC indicates North Atlantic Central Water whose $T$-$S$ characteristics reside along a straight line of positive slope, sketched in the T-S diagram of Fig. 4A by the double-headed arrow. The fractions $\alpha$ are normalized (they add up to

1) and change with time, the most notorious changes taking place at tidal timescale when $\alpha_{NAC}$ may reach high values. The changes of the fractions in the subsets of densest samples, on which we focus, are much more reduced. They must be proportional to the flows per unit width through CSN and CSS, thus inducing changes in $X_{ES}$ even if $X_{CSN}$ or $X_{CSS}$ do not




change. Therefore the observations at ESP are potentially useful to investigate the out-of-phase fluctuations of the flows across CSN and CSS illustrated in Fig. 5C, which allows extending our field information back to year 2004 when the monitoring station in ESP was first deployed.

The temperature difference between the Mediterranean waters and the NACW they mix with is much less than the salinity

difference (Fig. 4A), which results in salinity changes greater than temperature changes at ES with regards to the Mediterranean values. Actually, about 88% of the density difference between the mean value of the densest samples at ESP and CSN (large filled grey and blue dots in the inset of Fig. 4A, respectively) is accounted for by the salinity difference and only 12% by the temperature change (72% and 28% in the case of ESP and CSS). A consequence is that the weak signals that the fluctuating flows across CSN and CSS can cause in the variables observed at ESP via Eq. (1) are expected to be

better preserved in the temperature series, a fact already noted in previous works (García-Lafuente et al., 2007, Naranjo et al., 2012). In fact, Appendix A shows that any salinity signal in the Mediterranean waters identifiable at CS is very probably erased at ESP if $\alpha_{NAC}$ in the mixing exceeds 1%, whereas temperature signals require a greater fraction of around 6-8% to be faded out. The analysis carried out in García-Lafuente et al. (2011) suggests that the fraction of NACW entrained by the deep Mediterranean outflow when it passes over ESP is 4%, implying that Mediterranean signals will be found more easily

in temperature rather than in salinity series. Therefore, the potential temperature series is used in the following analysis.

Figure 6A shows the potential temperature series of the maximum density samples at ESP filtered with a Gaussian filter of 21 day$^{-1}$ cut-off frequency to remove the fortnightly variability and enhance the weaker subinertial signals. It is divided in two subseries, plotted in different colours, because the station did not register during part of years 2011 and 2012. Figure 6B shows the proxy of the WAG strength processed in the same way as the temperature series. The visual comparison of both

panels suggests that maxima in potential temperature tend to happen when the height of the WAG shows minima, which is the expected behaviour in our hypothesis: a minimum WAG index correspond to weak anticyclonic gyres, which in turn facilitates the evacuation of LIW and other intermediate waters (more flow across CSN) and the ensuing increase of temperature in the mixed water at ESP. The opposite would happen in case of maximum WAG index, which would correspond with minima of potential temperature at ESP.

Figure 6C shows the lagged correlation of both variables using the colour code of Fig. 6A and 6B (see caption). The negative values confirm the anti-correlation pointed out but it also shows that the maximum negative correlation does not happen at lag 0. In the case of the first piece (blue curves) the minimum R=-0.41, which is significant at 95% confidence level, occurs at lag +6 days, the temperature lagging the WAG index. Such delay is consistent with a cause-effect relationship if the WAG fluctuations cause the temperature changes at ESP. García-Lafuente et al. (2009), applying an empirical orthogonal function

analysis to altimetry data, found a similar delayed response in their study of the drainage of WMDW from the Alborán Sea. The correlation diminishes to R=-0.38 for the second and shorter piece (red curves), and peaks at lag +29 days, which is an exaggerated delay that does not make much physical sense. Considering both pieces together, the correlation drops a bit more (R=-0.35) and the lag decreases to an intermediate value of +18 days (grey curves). Nevertheless, it still appears as an unrealistic lag that stems from the big delay found in the second piece.



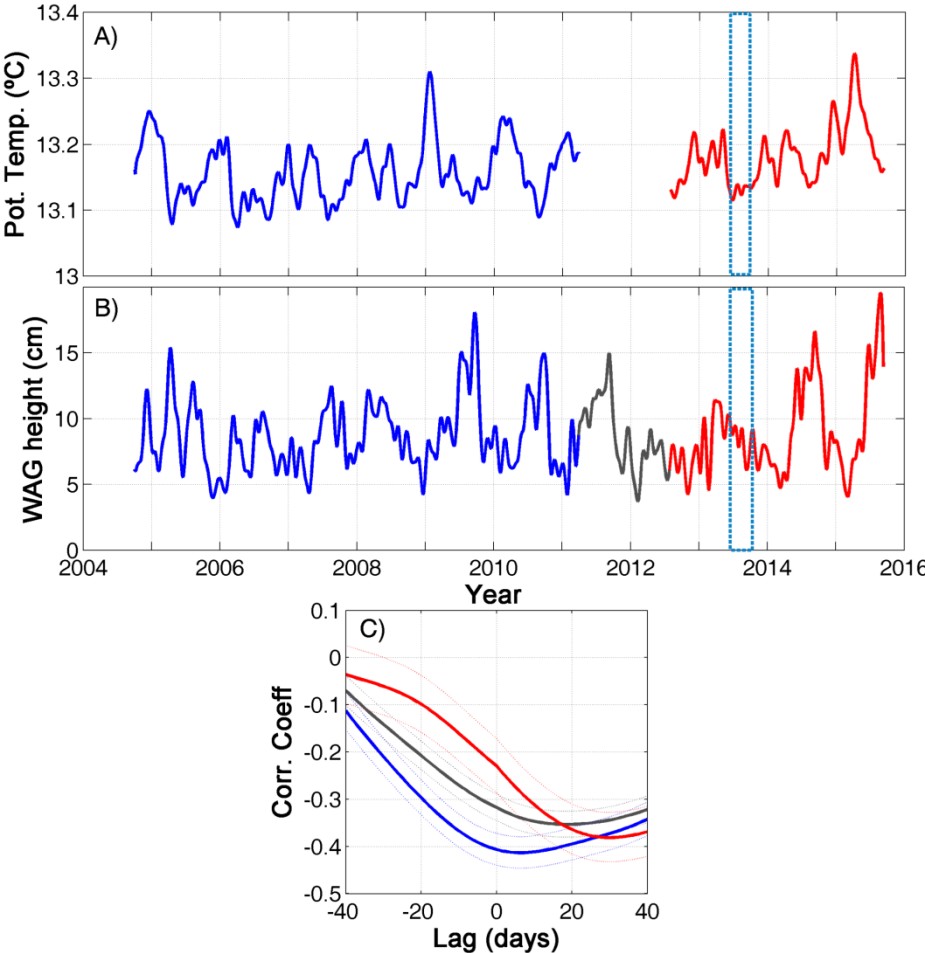

**Figure 6: A) Low-passed series (21-day$^{-1}$ cut-off frequency) of the potential temperature corresponding to the maximum density samples observed each semidiurnal tidal cycle at ESP. The series has been coloured in blue and red before and after the gap of years 2011-12, respectively. B) Low-passed series (same filter as in A) of the WAG height, the proxy used to characterize its strength. The same colour code has been applied to the periods of the subseries in A), whereas the period of the gap is shown in grey. C) Lagged correlation between the first (blue), second (red) and whole (black) series of WAG height and potential temperature at ESP. Dashed lines indicate the 95% confidence interval for R; positive lags mean that the first series (height of the WAG) leads the second one (potential temperature). The three-month period when the sampling at CS took place is indicated by the blue-dashed rectangles.**



## 4. Discussion and conclusions

The hypothesis that the composition of the Mediterranean outflow through the Strait of Gibraltar may be controlled by the fluctuations of the mesoscale anticyclonic gyre in the Western Alborán Sea has been tested using in situ observations and altimetry data for two different scenarios.

The first one encompasses an extensive spatial sampling of the flow through the main sills of the Strait, whose main weakness is the short duration of the collected time series. The comparison of the strength of the WAG, defined by a proxy consisting of the mean difference between its centre and its rim, with the flows through the north and south channels of CS suggests that the hypothesis is partially fulfilled since the weakening of the WAG coincides with an increase of the flow through CSN with regards to CSS (Fig. 5). However, the relationship is not accomplished or it is poor in other situations

(Fig. 5), which precludes to drawn definitive conclusions. The reason we put forward for this ambiguity is the short length of the series (around three months) that hardly covers six fortnightly cycles. The locally generated variability associated with these cycles overcomes the expectedly weaker signals that could come from the Mediterranean modulated by the WAG fluctuations. Filtering to remove the fortnightly cycle from such a short series would leave too few degrees of freedom to get a conclusion reasonably supported. Therefore, despite its partial support to the hypothesis, this first short length scenario

does not resolve the issue.

The second scenario involves the almost ten year long time series collected at Espartel sill by a monitoring station of the Mediterranean outflow. We argued that only the potential temperature series (Fig. 6A) gathers suitable conditions to search for signals of Mediterranean origin. The series available is divided in two pieces due to technical problems during years 2011-2012 (Sammartino et al., 2015). The first and longer piece provides an encouraging correlation of the correct sign (R=-

0.41) between potential temperature at ESP and the WAG proxy at a lag of 6 days, compatible with a cause-effect relationship under our hypothesis. With regards to this piece, Fig. 6B suggests an annual pattern for the WAG, the winter being the season of weaker gyres (local minima around the beginning of the years) and the summer and early autumn the time of better developed and robust WAGs. This seasonal pattern has been reported previously in the literature (Vargas Yáñez et al., 2002; Naranjo et al., 2012). Therefore and according to our hypothesis, winter would be the time when LIW is

more abundantly evacuated, leaving a warmer signature in the potential temperature at ESP, while summer and early autumn would favour the drainage of WMDW, which would cause a descent of the potential temperature. In general terms, this is what Fig. 6A shows up.

The second piece, however, shows a lower correlation (R=-0.38) that peaks at an unrealistic lag of 29 days.  A closer inspection to this piece reveals that the expected mirror-like behaviour between both variables depicted above is

satisfactorily achieved from the end of winter of year 2014 onwards, but that it is feebly met before that time. As seen in Fig. 6B, the seasonal tendency of the WAG to form stronger gyres during late summer and early autumn is not accomplished during years 2012 and 2013, a fact that could be behind the poorer correlation of both variables by the beginning of this second piece if we bear in mind that this seasonality appears to be the main driver of the correlation.





As mentioned in the Introduction, there are other possible causes or processes not necessarily correlated with the WAG fluctuations that can leave a signature in $T_{ESP}$. One of them is the winter formation of very large volumes of denser-than-average WMDW, which uplifts old WMDW in the vicinity of the Strait and facilitate its drainage. The two most prominent minima occurring in the years of 2005 and 2006 in Fig. 6A have this origin, according to García-Lafuente et al. (2007), and they did not need the assistance of the WAG to be achieved. Even more, they occurred in late winter / early spring, and not in summer, which is the season of the expected minimum $T_{ESP}$ according to our hypothesis. Another cause can be the interannual variability of the Mediterranean waters themselves, LIW and WIW on one hand, and WMDW on the other. CTD casts collected in the Strait in year 2013 showed a LIW colder than usual (Naranjo et al., 2015). This fact, along with the aforementioned absence of the expectedly strong summer WAG this year results in an unusual pattern and a reduced correlation between WAG and $T_{ESP}$ during the time of the field experiment (see dashed blue rectangles in Fig. 6A, 6B), which is an unfortunate coincidence. Notwithstanding, during this period the WAG index displays a tendency to diminish that corresponds with a slight increase of $T_{ESP}$, in agreement with out-of-phase behaviour of both variables expected under our hypothesis. Finally, another source of deviation may be the interannual variability of the properties of the NACW the Mediterranean waters mixes with (García-Lafuente et al., 2015). Should it be warmer than usual, the temperature signals coming from the Mediterranean Sea may not be identifiable. Such situation could be met, for instance, if a fraction $\alpha_{NAC} \geq 0.05$ of NACW at 15°C mixes with Mediterranean waters previously mixed at a ratio $r \geq 1$ (see Fig. A2 in appendix A). The resulting mixed water would probably lie in the zone labelled "No Mediterranean signal in $T_{ESP}$" in Fig. A2 of appendix A, and the correlation with the WAG fluctuations would be lost.

Despite these limitations, the reasonably good correlation found between the potential temperature at ESP and the proxy of the WAG is an encouraging result if we take into account the completely different and fully independent nature of both series. Therefore we conclude that the presence of a robust WAG helps ventilate the WMDW and that the evacuation of this water is preferably achieved during the periods of stable and well-developed gyres, which usually coincide with summer-early autumn season. On the contrary, wintertime, the season of weaker or, even, absent WAGs, favours the evacuation of the LIW. Due to the weakness of the signal and to some observed discrepancies, more research is needed to corroborate this hypothesis, which should include numerical modelling. The very subtlety of the mechanisms and processes involved and the complexity of the Strait hydrodynamics anticipate the use of models that likely are beyond the state of the art of the models presently in use.

### Acknowledgements

Data from the monitoring station deployed at Espartel Sill were collected in the frame of the Spanish Government funded "INGRES" projects (REN2003-01608, CTM2006-02326, CTM2009-05810/MAR and CTM2010-21229-C02-01/MAR). This monitoring station also contributes to the Mediterranean Sea monitoring network of the HYDROCHANGES program sponsored by the CIESM. The twin mooring lines deployed at Camarinal Sill in summer of 2013 are part of the experimental effort carried out within the Excellence Project MOCBASE (PE12-RNM-1540) funded by the Regional Government of





Junta de Andalucía. CN acknowledges a research contract associated with the Spanish funded project CTM2013-40886P. JCSG was supported by the Spanish Ministerio de Economía y Competitividad under the research contract JCI-2012-13451 of the "Juan de la Cierva" program. SS acknowledges a post-doc contract of Junta de Andalucía linked to MOCBASE project. We also acknowledge the Copernicus Marine Environment Monitoring Service (CMEMS) for making the altimetry

data used in this study freely available. This is the publication XXX from CEIMAR Publication Series.

**Appendix A.**

In reference to the inset of Fig. 4A, the dispersion of dots around the mean values (written inside brackets hereinafter, $\langle \cdots \rangle$) of CSN or CSS waters reflects their variability, which eventually will include Mediterranean signals advected by the outflow. The standard deviation (std) is a good metric of such dispersion. If the mean water properties at ESP $\langle T_{ESP} \rangle$ and

$\langle S_{ESP} \rangle$, are farther than $n$ times the std (the choice $n=3$ is adequate) from the mean value of the closest Mediterranean water (either CSN or CSS), it is reasonable to conclude that any Mediterranean signal that could have been advected by the outflow will have faded out in the observations at ESP due to mixing. This would happen whenever the fraction of NACW in the mixing exceeds a threshold value, which may differ if we deal with salinity or temperature.

During the field experiment analysed in this paper, the distance from $\langle T_{ESP} \rangle$ to $\langle T_{CSN} \rangle$ (the closest water to ESP for

temperature, see inset in Fig. 4A) is 0.047ºC, whereas 3×std($T_{CSN}$)=0.065ºC. According to our reasoning, temperature signals in the Mediterranean waters have a chance for being detected in $T_{ESP}$ series. In case of salinity, the distance from $\langle S_{ESP} \rangle$ to $\langle S_{CSS} \rangle$ (the closest water now) is 0.080, whilst 3×std($T_{CSS}$)=0.016, which hardly is one fifth of the required value. Salinity is far from fulfilling the condition, so that salinity signals in the Mediterranean waters will almost certainly disappear in $S_{ESP}$ series. Temperature is the right series to investigate Mediterranean signals in ESP. Although the previous numerical values

are for the particular period when the dataset was acquired and might not be valid for other periods, the conclusion that Mediterranean signals are better preserved in temperature than in salinity at ESP still holds because of the much greater difference in salinity than in temperature between the NACW and the Mediterranean waters.

The fraction of NACW in the mixed water at ESP necessary to erode the weak signals that could be in the Mediterranean waters is estimated now. Figure A1 sketches the mixing between NACW, whose specific properties change along the

double-headed arrow, and the underlying Mediterranean waters of characteristics CSN and CSS. The dots in the different outlined lines indicate possible results of mixing with different intervening fractions $\alpha_{CSN}$, $\alpha_{CSS}$, and $\alpha_{NAC}$. The particular case we focus on is represented by the large empty dot (labelled ESP), which lies in the mixing line of thick dots. Obviously, the position of this dot in the TS diagram depends on the fractions participating in the mixing as well as on the assumed values for CSN and CSS.

The normalization condition

$$\alpha_{CSN} + \alpha_{CSS} + \alpha_{NAC} = 1 \tag{A.1}$$





must be fulfilled in addition to eq. [1]. Our interest is the NACW fraction $\alpha_{NAC}$. Assuming that the mean properties of CSN and CSS ($<X_{CSN}>$, $<X_{CSS}>$, $X$ representing $T$ or $S$) are known, Eq. (1) and (A.1) still contain five unknowns: the three fractions, $X_{ESP}$ and $X_{NAC}$. A new relationship between fractions $\alpha_{CSN}$ and $\alpha_{CSS}$ can be established if we see the final product $X_{ESP}$ as the result of a first mixing between CSN and CSS to give point P, followed by a subsequent mixing between P and

5    the NACW. The ratio of the fractions $\alpha_{CSN}/\alpha_{CSS}$ is inversely proportional to the ratio of the distances $d_{CN}$ and $d_{CS}$ (see inset of Fig. A1)

$$\frac{\alpha_{CNS}}{\alpha_{CSS}} = \frac{d_{CS}}{d_{CN}} = r$$

(A.2)

The ratio $r$ varies from 0 in which case point $P$ coincides with CSS (and $d_{CS}=0$) to $\infty$ if $\alpha_{CSS}=0$ (point $P$ moves to CSN and $d_{CN}$ becomes 0). The interval $0.25 < r < 4$ ($\alpha_{CSS}=4\alpha_{CSN}$ to $\alpha_{CSN}=4\alpha_{CSS}$) contains all the reasonable values of $r$ in practice.

10   Eq. (1),([A.1), and (A.2) can be solved for the NACW fraction $\alpha_{NAC}$ to give:

$$\alpha_{NAC} = \frac{r(X_{ESP} - <X_{CSN}>) + (X_{ESP} - <X_{CSS}>)}{r(X_{NAC} - <X_{CSN}>) + (X_{NAC} - <X_{CSS}>)}$$

(A.3)

which contains three unknowns ($X_{ESP}$, $X_{NAC}$, $r$) as long as $<X_{CSN}>$ and $<X_{CSS}>$ are known.

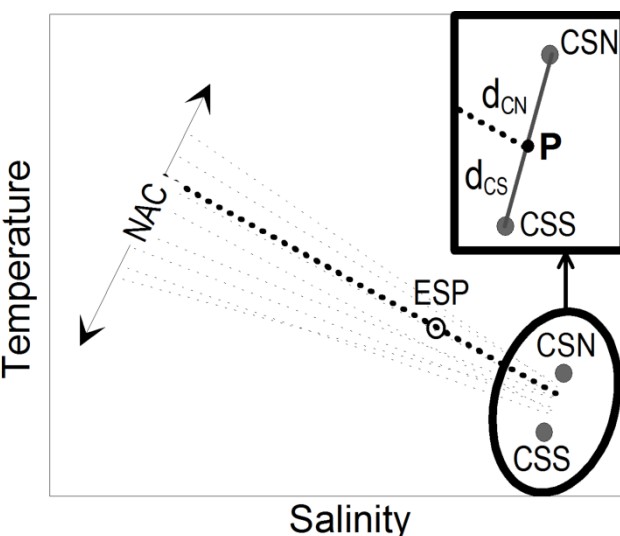

**Figure A1: Sketch of the mixing between two Mediterranean waters of characteristics CSN and CSS (large filled dots) and the NACW water mass (double headed arrow), based on Figure 4A (dimensions exaggerated). Several possible mixing lines with different fractions $\alpha_{CSN}$, $\alpha_{CSS}$, and $\alpha_{NAC}$ have been sketched. The particular case of mixed water with characteristics ESP (empty dot), whose position in the TS diagram depends on the fractions intervening in the mixing, is discussed in the text. This empty dot**
20  **can be seen as the result of a mixing between CSN and CSS to give point P (see inset), followed by a subsequent mixing between P and the NACW.**



We will focus on the period of the field experiment and identify these means with the mean values displayed in the inset of Fig. 4A. Figure A2 shows contours of $\alpha_{NAC}$ as a function of $X_{ESP}$ and $X_{NAC}$ for three different values of $r$, and for both $X=T$ and $X=S$. While the fraction $\alpha_{NAC}$ exhibits a small dependence on $r$ for low ratios in the temperature analysis (upper panel of

Fig. A2), it is insensitive to $r$ in the salinity case. The mean values $<T_{ESP}>$ and $<S_{ESP}>$ observed at ESP during this period are plotted as dotted green lines, and the aforementioned threshold values as thick grey lines ($T^*=<T_{CSN}>+3\times std(T_{CSN})=13.15°C$, $S^*=<S_{CSS}>-3\times std(S_{CSS})=38.47$). The zones where signals in the Mediterranean waters could still be detected in the corresponding series at ESP for this particular period of time are indicated. Even when the criterion $3\times std$ is arguable, the position of the observed $<S_{ESP}>$ with regards to $S^*$ suggests that the probability of detecting Mediterranean signals in salinity

at ESP is very low, whereas it is possible to find them in temperature.

Notice that both panels in Fig. A2 indicate a similar fraction $\alpha_{NAC}$ of the order of 3.5 to 5.5 % in the mixed water at ESP for typical values of NACW ($T\sim14.5°C$; $S\sim36.0$), in agreement with previous estimates (García-Lafuente et al., 2007, 2011). This value is significantly greater than the $\sim1\%$ deduced from the lower panel of Fig. A2 in order to detect signals in $S_{ESP}$. Obviously, the result of the analysis depends on the mean values of temperature and salinity at CSN and CSS ($<X_{CSN}>$ and

$<X_{CSS}>$), which change with time. For the expected fluctuations of these variables and the observed fractions of $\alpha_{NAC}$, however, the possibility that the salinity observed at ESP provides useful information about the Mediterranean signals carried by the outflow is very unlikely.

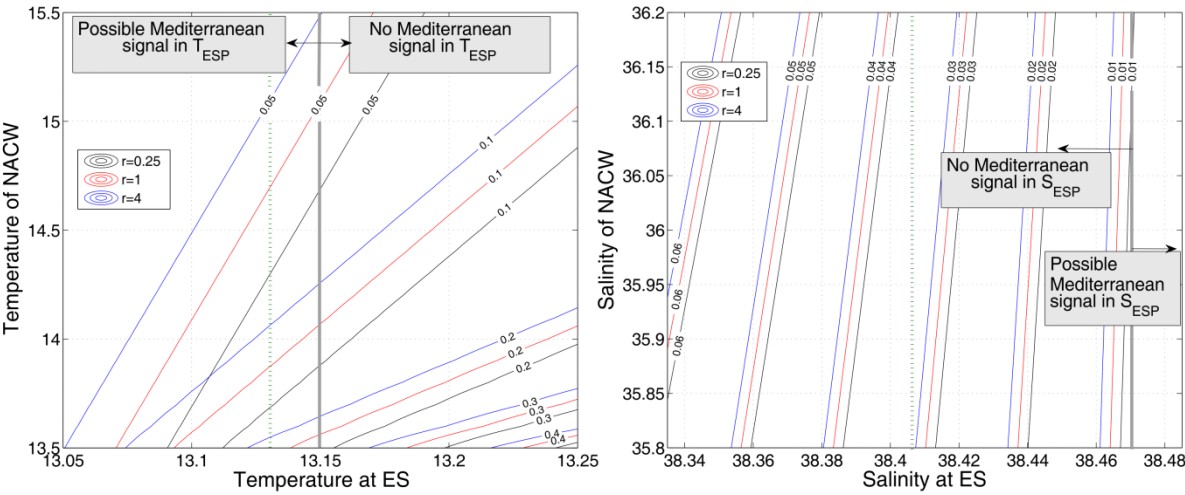

**Figure A2: Left: contours of the fraction $\alpha_{NAC}$ in the mixing (Eq. A.3) as a function of NACW and final mixed water at ESP temperatures for 3 different values of the ratio r (see legend; see also text for details). The green dashed line is the mean temperature at ESP in our dataset, whereas the thick solid grey line represents the "threshold" value for keeping chances of detecting, at least partially, the weak Mediterranean signals in the outflow at ESP. Right: same as left panel except for salinity.**



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
