# Peer review of "The Mediterranean Outflow in the Strait of Gibraltar and its connection with upstream conditions in the Alborán Sea."

_Ocean Science, 2016_

## Referee Comment (RC2)

Review of the paper entitle:

"*The Mediterranean Outflow in the Strait of Gibraltar and its connection with upstream conditions in the Alborán Sea*" by Jesús García-Lafuente, Cristina Naranjo, Simone Sammartino, José C. Sánchez-Garrido, Javier Delgado

The manuscript tackle an important and relevant scientific issue regarding the analysis of observations downstream and upstream the Gibraltar Strait; in particular in four different sites (Espartel Sill (ESP), Camarinal Sill North (CSN), Camarinal Sill South (CSS) and in the Western Alboran Gyre (WAG) through the proxy (deduced by altimetry data), in order to evaluate a connection between the strength of the WAG and the hydrological composition of the Mediterranean Outflow (MO).

This study is very interesting because increase understanding on the dependence expected between the variability of the two different ocean regions that the Gibraltar Strait connects (Gulf of Cadiz-Tangeri Basin (TB) and Alboran Basin respectively).

The analysis of the connection between the two side of the Strait is based on the study of the physical mechanisms that drive the variability of the MO; focusing on the main ocean phenomena observed in Gibraltar Strait, like mixing between different types of water of Mediterranean and Atlantic origin. The present paper is also relevant for ocean climate variability studies and in particular for research and simulation of the interaction between Mediterranean Sea and North Atlantic ocean.

For all these reasons that the results of this paper are very interesting for the oceanographic communities and in particular for those scientists more implicated on the Mediterranean-Atlantic interaction or on its parameterization in numerical climate models. Therefore I recommend this paper to be published, almost in the present version, however I would like to give some general comments and minor revisions in order to increase the impact of this manuscript around the oceanographic (and not only) community.

**General comments:**

The scientific matter of the manuscript isn't a really new augment, actually in the literature there are many example on this, either in the modelling field or like in this case in the analysis of the *in situ* observations. But the novelty of this manuscript (at least in my knowledge) is on the very detailed analysis of the hydrological characteristic in the key points, like CSN and CSS and ESP, crucial on determine the interaction, at different scale, between Mediterranean Sea and the Atlantic ocean.

However I have some doubts on the efficacy of the proxy (the altimetry height) used for analyse the WAG strength and its variability. I think that the matter of this manuscript is a little bit more complex.
Is matter of fact that the inflow/outflow is regulated not by a "single gyre" (WAG) but from the "double gyre". Actually, from satellite observations and from numerical simulations (see references 2 to 6 below), the Alboran Sea is dominated by the `double gyre' system and the Almeria-Oran front could be a good "proxy" of the variability of this system and consequently of the variability of the Atlantic water enters the Alboran via the straits of Gibraltar. Long-term monitoring of these currents is possible using data from satellite altimetry and finally will represent better the complexity of the processes of which the authors of this manuscript want to connect at the variability of MO.

Moreover, correctly in the manuscript the authors investigate in details on the mixing that take place inside and outside the Gibraltar Strait and the Almeria-Oran front again is also a proxy of these events, because is link also to the hydraulic condition along the Strait. The authors know very well (they wrote many papers on that) that the hydraulic control produces a hydraulic jump (forced by the tide) and consequently more vertical mixing is enhanced. These events in my opinion can modulate also the variability of the MO at larger frequency respect to the tide.

The second suggestion that I would like to put on the table for further discussion regard the Gulf of Cadiz, everybody know that when the MO pass over ESP has lost its original characteristic, taking now the properties of the source water of the MW that will be later observed in the North Atlantic, but still maintaining the memory of the originated Mediterranean water, in fact, following Fig. 4 of Fusco et al, 2008, is very evident the impact on the MO hydrological value of the quasi-periodical extraction and evacuation of WMDW from the Mediterranean into the Atlantic, that is the matter of this manuscript. Therefore, should be very interesting to verify the hydrological characteristic of MO in the Gulf of Cadiz and its interannual variability in relation of those observed in CSN, CSS and ESP.

**Minor revision:**

Fig.2 the range of the value of density is wrong;

Line 30-31 page 10 and line 1-3 page 11 I suggest to cut this sentence, the difference of the flow is so small that don't permit further considerations;

Fig.6 using this data set could be very interesting to do a SSA (Singular Spectral Analysis) of the time series, in order to capture the main low-frequency variability of this time series and verify the occurrence of a ghost limit cycle related to a physical oscillation of the dynamical system that has generated the time series (Ghil et al. 2002).

**References.**

1. Fusco, G., Artale V., Cotroneo Y.; Thermohaline variability of Mediterranean Water in the Gulf of Cadiz over the last decades (1948-1999), Deep Sea Research Part I: Oceanographic Research Papers, Volume 55, Issue 12, Pages 1624-1638, 2008;
2. Alvaro Peliz, Dmitri Boutov, Ana Teles-Machado. (2013) The Alboran Sea mesoscale in a long term high resolution simulation: Statistical analysis. Ocean Modelling 72, 32-52.
3. Speich, S., G. Madec, and M. Crépon, 1996: A strait outflow circulation process study: The case of the Alboran Sea. J. Phys. Oceanogr.,26, 320–340;
4. Whitehead, J. A., and A. R. Miller, 1979: Laboratory simulation of the gyre in the Alboran Sea. J. Geophys. Res.,84, 3733–3742.
5. Helen M. Snaith, S. G. Alderson, J. T. Allen and T. H. Guymer, Monitoring the eastern Alboran Sea using combined altimetry and in situ data, Phil. Trans. R. Soc. Lond. A (2003) 361, 65-70;
6. Lanoix, F. 1974, Projet Alboran, etude hydrologique et dynamique de la mer d' Alboran. Technical report no. 66. Brussels: NATO.
7. Ghil, M., and Coauthors, 2002: Advanced spectral methods for climatic time series. Rev. Geophys., 40, 1003, doi:10.1029/2000RG000092.

---

## Referee Comment (RC1) · Anonymous Referee #1 · 19 Jan 2017

General comments The main goal of this paper is the study of the possible connection between the composition of the Mediterranean outflow and the strength of the Gyre in the western Alborán Sea. The authors' hypothesis is that the stronger presence in the outflow either of the Western Mediterranean Deep Water or of the Levantine Intermediate Water is, respectively, correlated with a stronger or a weaker gyre. To detect the presence of the water masses and their fluctuations, they used two sets of in situ data, a shorter set (about 3 months) obtained with moored lines in the Camarinal Sill channels, and a historical long-term (almost 10 years long) dataset obtained at Espartel Sill. To estimate the strength of the gyre, sea level anomaly data were used. In general, the text is clear and is well complemented by the figures. As the subject of the

Mediterranean outflow composition and respective time variability is still not completely solved, all the contributions helping to have a better perception of this subject, especially those grounded on in situ data, are quite valuable and worth to be published, as happens with the present results, after some minor corrections.

Specific comments - There is evidence of seasonal variability in the Mediterranean outflow outside of Gibraltar, namely in the Gulf of Cadiz, as reported in previous published work. Could any link be established between the process identified in the present work and the seasonal fluctuations of the outflow farther downstream? Please comment on that. - What is the relative importance for the Mediterranean Outflow composition of the Alborán Gyre influence as compared to the influence of the annual process of the Western Mediterranean Deep Water formation (e.g., Garcia-Lafuente et al., 2007)? - Page 10, lines 20 - 23: clarify the paragraph starting with "These fluctuations ..." (too many options in brackets). Perhaps a clearer version could be: These fluctuations are explained in terms of local enhanced (reduced) mixing with the overlying NACW and, to a lesser extent, with higher (lower) WMDW aspiration driven by the enhanced (reduced) tidal currents over the sill in spring (neap) tides.

Technical corrections Page 1, line 25: that include years Page 1, line 25: Vargas-Yáñez et al., Page 2, line 8: García-Lafuente et al., Page 3, line 5: with regard to CSN Page 3, line 11: Sammartino et al. (2015) Page 3, line 12: has been analysed Page 5, line 12: with regard to CS Page 6, line 12: Wesson and Gregg, 1994 Page 9, line 2: should specify which interface Page 11, line 11: of July and of August Page 11, line 32: inducing changes in X ESP Page 12, line 5: changes in ESP with regard to Page 12, line 8: by the temperature difference Page 12, line 23: The opposite would happen or actually happens? Page 12, line 25: lagged correlation (R) Page 13, line 7: Lagged correlation (R) Page 14, line 7: mean difference in SLA Page 14, line 12: the expected weaker Page 14, line 20: proxy with a 6-day lag Page 14, line 21: with regard to this piece Page 15, line 3: which uplift Page 16, line 9: properties at ESP, Page 16, line 17:
3xstd(SCSS)= Page 17, line 10: Eq. (1), (A.1) and (A.2) Page 17, Figure A1: should be NACW instead of NAC Page 18, line 9: with regard to S\* Page 18, Figure A2: in the temperature and salinity axes: Temperature (Salinity) at ESP

---

## Author Comment (AC1) · 16 Feb 2017

Anonymous Referee #1 General comments: The main goal of this paper is the study of the possible connection between the composition of the Mediterranean outflow and the strength of the Gyre in the western Alborán Sea. The authors' hypothesis is that the stronger presence in the outflow either of the Western Mediterranean Deep Water or of the Levantine Inter- mediate Water is, respectively, correlated with a stronger or a weaker gyre. To detect the presence of the water masses and their fluctuations, they used two sets of in situ data, a shorter set

(about 3 months) obtained with moored lines in the Camarinal Sill channels, and a historical long-term (almost 10 years long) dataset obtained at Espartel Sill. To estimate the strength of the gyre, sea level anomaly data were used. In general, the text is clear and is well complemented by the figures. As the subject of the Mediterranean outflow composition and respective time variability is still not completely solved, all the contributions helping to have a better perception of this subject, especially those grounded on in situ data, are quite valuable and worth to be published, as happens with the present results, after some minor corrections.

Thank you for your opinion and the points raised in your review, which we address below.

Specific comments - There is evidence of seasonal variability in the Mediterranean outflow outside of Gibraltar, namely in the Gulf of Cadiz, as reported in previous published work. Could any link be established between the process identified in the present work and the seasonal fluctuations of the outflow farther downstream? Please comment on that.

Yes. There is. Recently, Bellanco et al., (2016) found a seasonal signal in the Gulf of Cádiz consisting of a cooler, saltier and denser MOW (Mediterranean Outflow Water) flowing in winter (March), which seems to coincide with the salinity signal in May-June showed previously in Fusco et al., (2008); authors attribute the signal to the maximum outflow, which occurs around this period (Sammartino, et al., 2015) and also to the less stratified Eastern North Atlantic Central Water (ENACW) overlying the Mediterranean Outflow Water (MOW). Both facts together result in an enhancement of the salt injection that finally originates the winter salinification of the MOW residing below 250m in the Gulf of Cádiz (Bellanco et al., 2016). Regarding the second part of your remark, the signal analyzed in this manuscript brings interannual variability to the MOW, whether or not these signals have an effect downstream in the Gulf of Cádiz. Strictly speaking, the answer to your point should be no in the sense that it is not possible to get an unquestionable conclusion about the suggested link with the data at hands. However,

the question remains open and deserving further study (which would imply a synoptic experiment like the one presented here but adding some moorings in the MOW path in the Gulf of Cádiz), so that in the new version we have made a short mention to this possible link in the Introduction, citing the abovementioned authors (lines 10-11, pag.3) and, again, in the Discussion (lines 27-32, pag.14) a bit more extensively.

- What is the relative importance for the Mediterranean Outflow composition of the Alborán Gyre influence as compared to the influence of the annual process of the Western Mediterranean Deep Water formation (e.g., Garcia-Lafuente et al., 2007)?

Certainly, besides the proposed influence of the Western Alborán Gyre strength there are other processes leaving a signal in the outflow water, one of them is the mentioned in García-Lafuente et al., (2007) as you commented. We had already made a discussion on the issue in page 15 (lines 7-13) and also in the Introduction (page 3, lines 24-30).

– Page 10, lines 20 – 23: clarify the paragraph starting with "These fluctuations . . ." (too many options in brackets). Perhaps a clearer version could be: These fluctuations are explained in terms of local enhanced (reduced) mixing with the overlying NACW and, to a lesser extent, with higher (lower) WMDW aspiration driven by the enhanced (reduced) tidal currents over the sill in spring (neap) tides.

Ok, we have now changed the paragraph as recommended, thanks for your suggestion.

Technical corrections Page 1, line 25: that include years Page 1, line 25: Vargas-Yáñez et al., Page 2, line 8: García-Lafuente et al., Page 3, line 5: with regard to CSN Page 3, line 11: Sammartino et al. (2015) Page 3, line 12: has been analysed Page 5, line 12: with regard to CS Page 6, line 12: Wesson and Gregg, 1994 Page 9, line 2: should specify which interface Page 11, line 11: of July and of August Page 11, line 32: inducing changes in X ESP Page 12, line 5: changes in ESP with regard to Page 12, line 8: by the temperature difference Page 12, line 23: The opposite would happen or actually happens? Page 12, line 25: lagged correlation (R) Page 13, line 7: Lagged

correlation (R) Page 14, line 7: mean difference in SLA Page 14, line 9: with regard to CSS Page 14, line 10: which precludes drawing Page 14, line 12: the expected weaker Page 14, line 20: proxy with a 6-day lag Page 14, line 21: with regard to this piece Page 15, line 3: which uplift Page 16, line 9: properties at ESP, Page 16, line 17 3xstd(SCSS)= Page 17, line 10: Eq. (1), (A.1) and (A.2) Page 17, Figure A1: should be NACW instead of NAC Page 18, line 9: with regard to S* Page 18, Figure A2: in the temperature and salinity axes: Temperature (Salinity) at ESP.

All technical corrections were modified in the text, thanks for the detailed revision and the suggestions.

Please find in the supplement file the formatted document of these comments and the new version of the manuscript with corrections highlighted.

Please also note the supplement to this comment:
http://www.ocean-sci-discuss.net/os-2016-90/os-2016-90-AC1-supplement.pdf

**Supplement:**

General comments: The main goal of this paper is the study of the possible connection between the composition of the Mediterranean outflow and the strength of the Gyre in the western Alborán Sea. The authors' hypothesis is that the stronger presence in the outflow either of the Western Mediterranean Deep Water or of the Levantine Inter- mediate Water is, respectively, correlated with a stronger or a weaker gyre. To detect the presence of the water masses and their fluctuations, they used two sets of in situ data, a shorter set (about 3 months) obtained with moored lines in the Camarinal Sill channels, and a historical long-term (almost 10 years long) dataset obtained at Espar- tel Sill. To estimate the strength of the gyre, sea level anomaly data were used. In general, the text is clear and is well complemented by the figures. As the subject of the Mediterranean outflow composition and respective time variability is still not completely solved, all the contributions helping to have a better perception of this subject, espe- cially those grounded on in situ data, are quite valuable and worth to be published, as happens with the present results, after some minor corrections.

*Thank you for your opinion and the points raised in your review, which we address below.*

Specific comments - There is evidence of seasonal variability in the Mediterranean out- flow outside of Gibraltar, namely in the Gulf of Cadiz, as reported in previous published work. Could any link be established between the process identified in the present work and the seasonal fluctuations of the outflow farther downstream? Please comment on that.

*Yes. There is. Recently, Bellanco et al., (2016) found a seasonal signal in the Gulf of Cádiz consisting of a cooler, saltier and denser MOW (Mediterranean Outflow Water) flowing in winter (March), which seems to coincide with the salinity signal in May-June showed previously in Fusco et al., (2008); authors*

*attribute the signal to the maximum outflow, which occurs around this period (Sammartino, et al., 2015) and also to the less stratified Eastern North Atlantic Central Water (ENACW) overlying the Mediterranean Outflow Water (MOW). Both facts together result in an enhancement of the salt injection that finally originates the winter salinification of the MOW residing below 250m in the Gulf of Cádiz (Bellanco et al., 2016).*

*Regarding the second part of your remark, the signal analyzed in this manuscript brings interannual variability to the MOW, whether or not these signals have an effect downstream in the Gulf of Cádiz. Strictly speaking, the answer to your point should be no in the sense that it is not possible to get an unquestionable conclusion about the suggested link with the data at hands. However, the question remains open and deserving further study (which would imply a synoptic experiment like the one presented here but adding some moorings in the MOW path in the Gulf of Cádiz), so that in the new version we have made a short mention to this possible link in the Introduction, citing the abovementioned authors (lines 10-11, pag.3) and, again, in the Discussion (lines 27-32, pag.14) a bit more extensively.*

 - What is the relative importance for the Mediterranean Outflow composition of the Alborán Gyre influence as compared to the influence of the annual process of the Western Mediterranean Deep Water formation (e.g., Garcia-Lafuente et al., 2007)?

*Certainly, besides the proposed influence of the Western Alborán Gyre strength there are other processes leaving a signal in the outflow water, one of them is the mentioned in García-Lafuente et al., (2007) as you commented. We had already made a discussion on the issue in page 15 (lines 7-13) and also in the Introduction (page 3, lines 24-30).*

– Page 10, lines 20 – 23: clarify the paragraph starting with "These fluctuations . . ." (too many options in brackets). Perhaps a clearer version could be: These fluctuations are explained in terms of local enhanced (reduced) mixing with the overlying NACW and, to a lesser extent, with higher (lower) WMDW aspiration driven by the enhanced (reduced) tidal currents over the sill in spring (neap) tides.

*Ok, we have now changed the paragraph as recommended, thanks for your suggestion.*

Technical corrections Page 1, line 25: that include years Page 1, line 25: Vargas-Yáñez et al., Page 2, line 8: García-Lafuente et al., Page 3, line 5: with regard to CSN Page 3, line 11:  Sammartino et al.  (2015) Page 3, line 12: has been analysed Page 5, line 12: with regard to CS Page 6, line 12: Wesson and Gregg, 1994 Page 9, line 2: should specify which interface Page 11, line 11: of July and of August Page 11, line 32: inducing changes in X ESP Page 12, line 5: changes in ESP with regard to Page 12, line 8: by the temperature difference Page 12, line 23: The opposite would happen or actually happens? Page 12, line 25: lagged correlation (R) Page 13, line 7: Lagged correlation (R) Page 14, line 7: mean difference in SLA Page 14, line 9: with regard to CSS Page 14, line 10: which precludes drawing Page 14, line 12: the expected weaker Page 14, line 20: proxy with a 6-day lag Page 14, line 21: with regard to this piece Page 15, line 3: which uplift Page 16, line 9: properties at ESP, Page 16, line 17 3xstd(SCSS)= Page 17, line 10: Eq. (1), (A.1) and (A.2) Page 17, Figure A1: should be NACW instead of NAC Page 18, line 9: with regard to S* Page 18, Figure A2: in the temperature and salinity axes: Temperature (Salinity) at ESP.

*All technical corrections were modified in the text, thanks for the detailed revision and the suggestions.*

*Please find below the manuscript with the highlighted corrections.*

[revised manuscript text omitted]

---

## Author Comment (AC2)

Review of the paper entitle:

"*The Mediterranean Outflow in the Strait of Gibraltar and its connection with upstream conditions in the Alborán Sea*" by Jesús García-Lafuente, Cristina Naranjo, Simone Sammartino, José C. Sánchez-Garrido, Javier Delgado

The manuscript tackle an important and relevant scientific issue regarding the analysis of observations downstream and upstream the Gibraltar Strait; in particular in four different sites (Espartel Sill (ESP), Camarinal Sill North (CSN), Camarinal Sill South (CSS) and in the Western Alboran Gyre (WAG) through the proxy (deduced by altimetry data), in order to evaluate a connection between the strength of the WAG and the hydrological composition of the Mediterranean Outflow (MO).

This study is very interesting because increase understanding on the dependence expected between the variability of the two different ocean regions that the Gibraltar Strait connects (Gulf of Cadiz-Tangeri Basin (TB) and Alboran Basin respectively).

The analysis of the connection between the two side of the Strait is based on the study of the physical mechanisms that drive the variability of the MO; focusing on the main ocean phenomena observed in Gibraltar Strait, like mixing between different types of water of Mediterranean and Atlantic origin. The present paper is also relevant for ocean climate variability studies and in particular for research and simulation of the interaction between Mediterranean Sea and North Atlantic ocean.

For all these reasons that the results of this paper are very interesting for the oceanographic communities and in particular for those scientists more implicated on the Mediterranean-Atlantic interaction or on its parameterization in numerical climate models. Therefore I recommend this paper to be published, almost in the present version, however I would like to give some general comments and minor revisions in order to increase the impact of this manuscript around the oceanographic (and not only) community.

*Thank you for your opinion and the points raised in your review, which we address below.*

**General comments:**

The scientific matter of the manuscript isn't a really new augment, actually in the literature there are many example on this, either in the modelling field or like in this case in the analysis of the *in situ* observations. But the novelty of this manuscript (at least in my knowledge) is on the very detailed analysis of the hydrological characteristic in the key points, like CSN and CSS and ESP, crucial on determine the interaction, at different scale, between Mediterranean Sea and the Atlantic ocean.

However I have some doubts on the efficacy of the proxy (the altimetry height) used for analyse the WAG strength and its variability. I think that the matter of this manuscript is a little bit more complex.
Is matter of fact that the inflow/outflow is regulated not by a "single gyre" (WAG) but from the "double gyre". Actually, from satellite observations and from numerical simulations (see references 2 to 6 below), the Alboran Sea is dominated by the `double gyre' system and the Almeria-Oran front could be a good "proxy" of the variability of this system and consequently of the variability of the Atlantic water enters the Alboran via the straits of Gibraltar. Long-term monitoring of these currents is possible using data from satellite altimetry and finally will represent better the complexity of the processes of which the authors of this manuscript want to connect at the variability of MO.

Moreover, correctly in the manuscript the authors investigate in details on the mixing that take place inside and outside the Gibraltar Strait and the Almeria-Oran front again is also a proxy of these events, because is link also to the hydraulic condition along the Strait.

*As far as we know, the likely relationship or link between the Almeria-Oran (AO) front and the hydraulic condition in the Strait, was first put forward by Garret (1996, "The role of the Strait of Gibraltar….." in Dynamics of Mediterranean straits and channels, CIESM Science Series nº2), and connects with the concept of an overmixed Mediterranean proposed previously by Bryden and Stommel (1984). As you know, this general relationship comes from a simplified conceptual model, yet sensible and enlightening, which, when applied to the actual ocean, shows some weaknesses. Numerical models suggest that most of the time the flow in Gibraltar is hydraulically controlled (except for tidally-induced control flooding, which happens in the short time-scale), but the presence of the AO front is not so regular and permanent, experiencing dilated periods of time during which the front is not present or, at least, not identifiable (the very Report by Lanoix mentioned in your reference list provides an illustrative example). Therefore, search for an external AO-front-based proxy is not the best choice to establish a reliable link with the Strait of Gibraltar hydraulics.*

*In any case the remark you pointed out is quite interesting since, as you mention, the complete Alborán Sea system includes the two anticyclonic (western, eastern) gyres and the AO-front at the eastern boundary. Expectedly, the whole system influences the exchange. However, when looking for a proxy that represents this system, we finally opted for the Western Alborán Gyre (WAG) for three main reasons:*

*1. Being the nearest structure to the Strait, the WAG is expected to have a greater influence on the exchange.*

*2. As repeatedly mentioned in the literature (Vargas-Yañez et al., 2002, for instance), the WAG is a quasi-permanent feature while the Eastern gyre is more elusive (Garcia-Lafuente, 1998). The WAG is identifiable almost all year round, although there are exceptions occurring during some periods when it collapses and disappears (Sanchez-Garrido, et al., 2013), exceptions that are of the greatest interest to our work.*

*3. Last but not least, choosing the WAG allows us to define a rather simple and intuitive proxy from altimetry, even if the WAG is weak, while including all the mesoscale structures of the Alboran Sea complicates the definition of a representative index for our purposes and would introduce uncertainty about the proxy utility.*

The authors know very well (they wrote many papers on that) that the hydraulic control produces a hydraulic jump (forced by the tide) and consequently more vertical mixing is enhanced. These events in my opinion can modulate also the variability of the MO at larger frequency respect to the tide.

*We are aware of the great tidal induced mixing in the hydraulic transitions within the Strait, a fact that has been already commented on the manuscript (see Figure 2 and comments on that in pages 5-6). You are right mentioning that this short time-scale mixing may induce low-frequency signals in the outflow if one of the water mass involved (it shall be NACW in our case) changes seasonally. Although very intuitive and, probably somewhat redundant, we have opted for modifying the text and including a short sentence (lines 20-25 in pag. 15 of the new version) to mention explicitly this process.*

The second suggestion that I would like to put on the table for further discussion regard the Gulf of Cadiz, everybody know that when the MO pass over ESP has lost its original characteristic, taking now the properties of the source water of the MW that will be later observed in the North Atlantic, but still maintaining the memory of the originated Mediterranean water, in fact, following Fig. 4 of Fusco et al, 2008, is very evident the impact on the MO hydrological value of the quasi-periodical extraction and evacuation of WMDW from the Mediterranean into the Atlantic, that is the matter of this manuscript. Therefore, should be very interesting to verify the

hydrological characteristic of MO in the Gulf of Cadiz and its interannual variability in relation of those observed in CSN, CSS and ESP.

*Additionally to Fusco et al. (2008), the variability of the Mediterranean outflow was recently revisited by Bellanco et al. (2016) using a collection of CTD casts collected from 2005 to 2015. Both, Fusco et al., and Bellanco et al., found a salinity signal below 250 m (May-June in Fusco et al., March in Bellanco et al.) that is the most outstanding seasonal signal of the Mediterranean outflow in the Gulf of Cádiz.*

*Despite being a remarkable suggestion, the possibility of verifying or concluding beyond any doubt that the seasonal signal of the MOW in the Gulf of Cadiz comes from the processes discussed in our work is not realistic with the available data. It rather appertains to the speculation world from which can be drawn down only after carrying on a synoptic field experiment that involves the simultaneous deployment of mooring lines along the MOW path and the sills of the Strait. Even though such experiment is not discarded for the future, presently its realization is beyond our possibilities. Notwithstanding, we think it interesting to mention this possible link, so that in the Introduction of the new version (lines 10-11, pag.3) we have made a short mention to it, citing the abovementioned authors, and we do it again in the Discussion section (lines 27-33, pag.14), a bit more extensively in this case.*

**Minor revision:**

Fig.2 the range of the value of density is wrong;

*Actually, Fig.2c shows sigma-theta, not density. We have revised the units and checked they are correct*

Line 30-31 page 10 and line 1-3 page 11 I suggest to cut this sentence, the difference of the flow is so small that don't permit further considerations;

*We are not sure to have understood the point. The flow difference is usually well-discernibly different from zero, confirming that the flow across the southern channel tends to be consistently larger than across the northern channel, a fact already mentioned in older works (Candela et al., 1989; Bryden et al., 1994). The interesting point is that a time comes when the difference changes sign (we admit that it is not much different from zero then), indicating that the tendency of the flow to be greater across the southern channel has been broken. And that this time coincides with a notable weakening, if not a collapse, of the WAG, what, under our hypothesis, gives a good chance to LIW for flowing out in greater volumes. Therefore this small difference, even if small, is critical for our analysis and the starting point of the discussion of the data. It is because these differences are observed that we can justify the importance of the WAG in the cross-structure of the flow through the Strait. The sentence cannot be removed.*

Fig.6 using this data set could be very interesting to do a SSA (Singular Spectral Analysis) of the time series, in order to capture the main low-frequency variability of this time series and verify the occurrence of a ghost limit cycle related to a physical oscillation of the dynamical system that has generated the time series (Ghil et al. 2002).

*Thanks for your suggestion, in fact the SSA is an interesting tool to obtain a clean signal in noisy time series and it also gives information about the different oscillatory components in the signal. We have applied the SSA to the original temperature series that was used to obtain Fig.6 and we have found that, even if the reconstructed series results in a smoother signal compared with what we show in Figure 6A, no difference in the interpretation of the results stems from the application of this spectral analysis. Thus we prefer to maintain the previous method used in the manuscript (that is, the simpler Gaussian filtering of the original series) to avoid complicating the methodology section without further improvement of the results.*

[Figure]

*Panel A) is the original Figure 6A of the manuscript, panel B) shows the result of applying a Singular Spectral Analysis to the original θ series (grey line). The figure shows the reconstruction of the series using up to 8 Reconstructed Components. Note that B) is similar to the original filtered signal of panel A), where a Gaussian filter has ben used. Actually, the unique difference is the smoothness of the series, but both processing are acceptable for the purpose that concern us (the same applies to the Sea Level Anomaly signal in Fig. 6B)*

*Please find below the manuscript with the highlighted corrections.*

**References.**

*Bellanco, M. J., and Sánchez-Leal, R. F.: Spatial distribution and intra-annual variability of water masses on the Eastern Gulf of Cadiz seabed, Continental Shelf Research, 128, 26-35, http://dx.doi.org/10.1016/j.csr.2016.09.001, 2016.*

*García Lafuente, J. G. a., Cano, N., Vargas, M., Rubín, J. P., and Hernández-Guerra, A.: Evolution of the Alboran Sea hydrographic structures during July 1993, Deep Sea Research Part I: Oceanographic Research Papers, 45, 39-65, http://dx.doi.org/10.1016/S0967-0637(97)00216-1, 1998.*

*Garrett, C.: The role of the Strait of Gibraltar in the evolution of Mediterranean water, properties and circulation, BULLETIN-INSTITUT OCEANOGRAPHIQUE MONACO-NUMERO SPECIAL-, 1-20, 1996.*

*Sánchez Garrido, J. C., García Lafuente, J., Álvarez Fanjul, E., Sotillo, M. G., and de los Santos, F. J.: What does cause the collapse of the Western Alboran Gyre? Results of an operational ocean model, Progress in Oceanography, 116, 142-153, http://dx.doi.org/10.1016/j.pocean.2013.07.002, 2013.*

*Vargas-Yáñez, M., Plaza, F., García-Lafuente, J., Sarhan, T., Vargas, J. M., and Vélez-Belchi, P.: About the seasonal variability of the Alboran Sea circulation, Journal of Marine Systems, 35, 229-248, http://dx.doi.org/10.1016/S0924-7963(02)00128-8, 2002.*

**References.**

1. Fusco, G., Artale V., Cotroneo Y.; Thermohaline variability of Mediterranean Water in the Gulf of Cadiz over the last decades (1948-1999), Deep Sea Research Part I: Oceanographic Research Papers, Volume 55, Issue 12, Pages 1624-1638, 2008;

2. Alvaro Peliz, Dmitri Boutov, Ana Teles-Machado. (2013) The Alboran Sea mesoscale in a long term high resolution simulation: Statistical analysis. Ocean Modelling 72, 32-52.

3. Speich, S., G. Madec, and M. Crépon, 1996: A strait outflow circulation process study: The case of the Alboran Sea. J. Phys. Oceanogr.,26, 320–340;

4. Whitehead, J. A., and A. R. Miller, 1979: Laboratory simulation of the gyre in the Alboran Sea. J. Geophys. Res.,84, 3733–3742.

5. Helen M. Snaith, S. G. Alderson, J. T. Allen and T. H. Guymer, Monitoring the eastern Alboran Sea using combined altimetry and in situ data, Phil. Trans. R. Soc. Lond. A (2003) 361, 65-70;

6. Lanoix, F. 1974, Projet Alboran, etude hydrologique et dynamique de la mer d' Alboran. Technical report no. 66. Brussels: NATO.

7. Ghil, M., and Coauthors, 2002: Advanced spectral methods for climatic time series. Rev. Geophys., 40, 1003, doi:10.1029/2000RG000092.

[revised manuscript text omitted]

---

## Author Response (AR2)

**Revised Submission**

Dear Editor,

Thank for your suggestions and corrections. We have included new paragraphs in the manuscript and changed the Figure 1 as well as corrected other minor issues. You can find the point-by-point response and the manuscript tracked changes below this lines.

Sincerely,

Cristina, (on behalf of all authors).

Topic Editor Decision: Publish subject to minor revisions (Editor review) (22 Feb 2017) by Dr. Mario Hoppema

Comments to the Author:

Dear Dr. Naranjo and co-authors,

Thank you for the revised submission of you manuscript. I am mostly satisfied with your corrections and modifications, but still have some few comments myself, which are listed below. Please prepare the final version of your manuscript providing for these comments and then the manuscript can be published in Ocean Science.

The comment by referee #2 starting with "However I have some doubts on the efficacy of the proxy (the altimetry height) used …". I appreciate your response to that, but am surprised not to find anything from that in the revised manuscript. Please correct this.

We did not find it necessary to modify the text because it appeared obvious that the most suitable structure for defining any proxy was the WAG. However, after reading your remark, we have re-thought the issue and decided to insert a short sentence in page 10 lines 1-6 justifying our choice.

There are so many water masses and other features with abbreviations in the text that it becomes quite confusing. I suggest to add a table with all abbreviations for clarity. You might also use the full names scattered in some parts of the manuscript for enhancing the readability.

We agree that there are too many acronyms. We have now eliminated all the acronyms of the water masses keeping only the acronyms for the geographical location of the moorings (ESP, CSS, CSN) and the Western Alborán Gyre (WAG), which are the four more cited in the text.

P3, L6 upward looking (instead of: uplooking), also at other places in the manuscript. ok

P7, L8 T-S diagram (also at P8, L2) ok

Paragr 2.2.3. Is there any reference to the method of determining the strength of the WAG, i.e. as an example in another gyre where similar things have been done? This would enhance the support for the method considerably.

Some references have been now included in page 10 lines 15-17.

Figure 1 Please add all geographic names mentioned in the paper.

I think Fig. 1 is not quite sufficient as what it is showing. Important regions are not included: It would also be useful to see the location of the WAG, and possibly the other gyre of the double gyre system. Please modify accordingly.

Ok. We have now enlarged the map extent in Figure 1, and we included Spain, Morocco and the Western Alborán Gyre labels. We are aware that other geographic names as "Tyrrhenian Sea", "Strait of Sicily" etc. are not included, but these locations are mentioned only in the introduction and they are not essential for the comprehension of the manuscript. Should these labels be included, a new figure would be needed, and we consider it unnecessary.

References

Kinder, T.H. and Bryden, 1990: add publisher ok

Naranjo et al 2014: add pages and 63(0) is wrong. ok

Thank you and best wishes

Mario Hoppema

[revised manuscript text omitted]